# TRPV4 Role in Neuropathic Pain Mechanisms in Rodents

**DOI:** 10.3390/antiox12010024

**Published:** 2022-12-22

**Authors:** Patrícia Rodrigues, Náthaly Andrighetto Ruviaro, Gabriela Trevisan

**Affiliations:** Graduated Program in Pharmacology, Federal University of Santa Maria (UFSM), Santa Maria 97105-900, RS, Brazil

**Keywords:** chronic pain, oxidative stress, TRP channels, allodynia, TRPV1, TRPA1

## Abstract

Neuropathic pain is a chronic pain caused by a disease or damage to the somatosensory nervous system. The knowledge about the complete mechanisms is incomplete, but the role of oxidative compounds has been evaluated. In this context, we highlight the transient potential receptor vanilloid 4 (TRPV4), a non-selective cation channel, that can be activated by oxidated compounds. In clinical trials, the TRPV4 antagonist (GSK2798745) has been well-tolerated in healthy volunteers. The TRPV4 activation by oxidative compounds, such as hydrogen peroxide (H_2_O_2_) and nitric oxide (NO), has been researched in neuropathic pain models. Thus, the modulation of TRPV4 activation by decreasing oxidated compounds could represent a new pharmacological approach for neuropathic pain treatment. Most models evaluated the TRPV4 using knockout mice, antagonist or antisense treatments and detected mechanical allodynia, hyposmotic solution-induced nociception and heat hyperalgesia, but this channel is not involved in cold allodynia. Only H_2_O_2_ and NO were evaluated as TRPV4 agonists, so one possible target to reduce neuropathic pain should focus on reducing these compounds. Therefore, this review outlines how the TRPV4 channel represents an innovative target to tackle neuropathic pain signaling in models induced by trauma, surgery, chemotherapy, cancer, diabetes and alcohol intake.

## 1. Neuropathic Pain

Pain can be categorized as acute or chronic pain, where chronic pain is defined by the International Association for the Study of Pain (IASP) as “pain that persists or recurs for more than 3 months”. This pain occurs in approximately 20% of the general population and negatively affects the quality of life, a common cause of disability and suffering in patients [1]. The last revision of the International Classification of Diseases (ICD-11) included a new system for the classification of chronic pain as primary and secondary. The first is detected in patients with migraine, fibromyalgia, or complex regional pain syndrome. This type of chronic pain cannot be explained by a complication of another chronic pain condition. In contrast, secondary chronic pain arises initially as a symptom of another disease, such as cancer pain and diabetic polyneuropathy [2].

Neuropathic pain is a frequent type of chronic pain, affecting 7–10% of people worldwide. However, the incidence of neuropathic pain probably will enhance with the growing number of diagnoses for diabetes mellitus, the aging population, and chemotherapy for cancer treatment. The IASP defines this form of pain as “pain caused by a lesion or disease of the somatosensory nervous system” [1]. Additionally, it is classified as central pain (in multiple sclerosis, central post-stroke pain, and spinal cord injury) or peripheral pain (detected in diabetic polyneuropathy, chemotherapeutic administration, and peripheral nerve trauma). Patients often describe neuropathic pain as paresthesia, spontaneous (pricking or burning), or evoked pain (such as allodynia and hyperalgesia to mechanical or thermal painful stimuli). Moreover, negative sensory symptoms may also be present in neuropathic pain, including sensory loss and numbness [3,4].

The current pharmacological treatment for neuropathic pain involves medicines that only control painful symptoms. Then, distinct analgesics can be used as first-line therapies, including gabapentinoids (pregabalin and gabapentin), tricyclic antidepressants, and duloxetine (selective serotonin and norepinephrine reuptake inhibitor antidepressant, SSNRI). Alternative solutions should be used to manage neuropathic pain, such as high-concentration capsaicin patches (a transient receptor potential vanilloid 1–TRPV1 agonist found in chili peppers) and tramadol (an opioid agonist that also can act as a SSNRI) [5], although these compounds have adverse effects and still show limited efficacy in controlling neuropathic pain [3,4].

In this view, studying the mechanisms involved in neuropathic pain development could help design better therapeutic options and, in some cases of neuropathic pain, propose a way to prevent it, such as the neuropathic pain caused by chemotherapeutic administration or diabetes. Different mechanisms may lead to neuropathic pain induction, such as altered ion channel expression and function when there is a nerve injury or a disease of the somatosensory system. In addition, oxidative compounds can also be involved in neuropathic pain development [6]. Then, this process may cause increased excitation of the peripheral and central nervous nociceptive neurons or a reduction in the inhibition of ascending afferent pathways. The superfamily of ion channels named transient receptor potential (TRP) acts as sensors of oxidative compounds at the plasma membrane and can amplify several signaling pathways [7]. Therefore, specific TRP modulators, such as agonists, antagonists, and endogenous modulators, have been identified and analyzed for future therapeutic applications to treat chronic diseases, such as neuropathic pain [3,4,8].

## 2. Transient Receptor Potential (TRP)

The TRP superfamily is a non-selective cation channel initially identified in the *Drosophila* fly species. These channels are presented in different cell types and tissues, such as epithelial, immune, and neuronal cells. In mammals, the TRP superfamily has 28 described members, which are divided into six subgroups according to channel structure: TRP Canonical (TRPC), TRP Vanilloid (TRPV), TRP Melastatin (TRPM), TRP Polycystin (TRPP), TRP Mucolipin (TRPML) and TRP Ankyrin (TRPA) [7]. TRP channels are activated by pharmacological ligands, temperature, pH, and mechanical force [9]. Additionally, inflammatory and oxidative compounds generated after tissue lesions can activate or sensitize these channels [10]. After activation, the TRP channels mediate the influx of different ions, such as sodium (Na^+^), calcium (Ca^2+^) and magnesium (Mg^2+^) [7].

TRP channels share a typical structure of six transmembrane regions with amino and carboxyl terminus in the cytoplasm [9]. The first member of the TRP channel family that has its structure completely elucidated was TRPV1, followed by TRPA1 and TRPV4 [11]. The structural elucidation helped the investigation of their involvement in pathologies, such as pain, chronic inflammation, fibrosis and edema [12]. Moreover, TRPA1, TRPV1 and TRPV4 channels co-localize in primary sensory neurons and are related to inflammatory and neuropathic pain development in pre-clinical models [8]. In this view, we point out the TRPV subfamily that plays an essential role in neuropathic pain, inflammation, immunity, neuronal development, diabetes, cardiovascular disease, and cancer [13].

Among the six family members, TRPV1–4 are thermosensitive, while TRPV5–6 are not sensitive to temperature [13]. TRPV1 is activated by capsaicin and noxious temperatures (≥43 °C) and is associated with neuropathic and inflammatory pain induction [13]. Capsaicin is the only drug approved for clinical use that binds to TRP channels (TRPV1) and can be used, as a second-line drug, for diabetic peripheral neuropathy treatment (Qutenza^®^). Additionally, clinical trials have demonstrated capsaicin efficacy for post-herpetic neuralgia, post-traumatic or post-surgical nerve injury, human immunodeficiency virus (HIV)-induced neuropathy and chronic painful chemotherapy. However, capsaicin can only be used for topical application (local pharmacological action) due to the side effects, such as erythema, pruritus, reduced heat sensitivity and pain [14]. Therefore, TRP channels still need to be investigated as new pharmacological targets to treat neuropathic pain.

In this view, TRPV4 has been widely studied and linked to many channelopathies, suggesting a broad expression pattern and versatile physiological function [15]. The TRPV4 gene mutations have generated osteoarthropathy, skeletal dysplasia and peripheral neuropathies, manifesting as a variable combination of skeletal, motor and neuronal symptoms, including pain [16]. Therefore, the TRPV4 channel is particularly interesting due to its involvement in neuropathic pain symptoms [8].

### Transient Receptor Potential Vanilloid 4 (TRPV4)

The TRPV4 channel structure contains six transmembrane regions (S1–S6), similar to other TRP proteins. The intracellular N-region is linked to the S1 region and includes six ankyrin domains, a proline-rich domain and a linker domain with two β-strands [17]. The TRP helix pore-forming loop allows the ionic flow and is located between S5 and S6 domains. Additionally, extending from the S6 transmembrane region is observed the C-terminal domain with a folding recognition domain (FRD), a TRP domain and a PDZ-binding domain [17]. These structure characteristics are essential for assembly and localization, possibly contributing to the activation and modulation of the TRPV4 function [18]. The X-ray crystallographic analyses identify that the symmetric tetramer architecture of TRPV4 is similar to other TRPV channels, including TRPV1, TRPV2 and TRPV6. However, the unique arrangement of the S1–S4 domain packing against the S5–S6 transmembrane domains differentiated this channel from the other TRPVs. Thus, clarifying the molecular and cellular mechanisms underlying the TRPV4 channel is fundamental to understanding channel function and designing effective therapies [19].

The TRPV4 channel is expressed in immune cells, such as macrophages, neutrophils, and dendritic cells. Moreover, this channel is also expressed in sensory neurons, glial cells, the spinal cord, cortical pyramidal neurons, the thalamus, and cerebellum basal nuclei [20]. Thus, this channel is associated with inflammatory diseases that affect the central and peripheral nervous system, such as osteoarthritis, atherosclerosis, cancer pain, and neuropathies [21].

TRPV4 can be activated by hyposmolarity, non-noxious heat (38 °C), hydrogen peroxide (H_2_O_2_), low pH (5.0), mechanical forces, and ultraviolet B-rays (UVB) radiation. Additionally, exogenous or endogenous chemical compounds can activate or block the TRPV4 channel. The TRPV4 endogenous agonists include arachidonic acid, 5′,6′-epoxyeicosatrienoic acid (5′,6′-EET) and dimethylallyl pyrophosphate (Figure 1). In addition, the exogenous agonists include x-3 polyunsaturated fatty acids, bisandrographolide A, 4α-phorbol 12,13-didecanoate (4α-PDD) and GSK1016790A [17]. Although the therapeutic potential of selective TRPV4 agonists has been hypothesized, most clinical interest has concentrated on channel inhibition [22]. In this context, TRPV4 antagonists such as HC-067047, RN-1734 and GSK2193874 have been evaluated and used in pre-clinical models to reduce nociception [22].

Furthermore, when TRPV4 is activated, Ca^2+^ influx enhances the signaling of protein kinase C (PKC)-dependent phosphorylation. The PKC activation in the dorsal root ganglion (DRG) neurons generates TRPV4 sensitization and plays a role in nociception [23]. Another signaling pathway that activates TRPV4 is protease activating receptor 2 (PAR2), probably by PKC and protein kinase A (PKA) activation. The PAR2 is a G protein-coupled receptor that is expressed in alveolar macrophages, endothelial cells, and epithelial cells. This receptor can modulate inflammatory responses due to pro-inflammatory cytokines production, such as interleukin-1β (IL-1β), interleukin-6 (IL-6) and interleukin-8 (IL-8), which are involved in pain control [18].

In addition, nitric oxide (NO), has been associated with mechanisms of neuropathic pain and peripheral nerve injury, possibly due to the modification of protein kinase and ion channels [24]. The TRPV4-mediated Ca^2+^ influx can activate the inducible nitric oxide synthase (iNOS)-nitric oxide, which increases NO release. NO can activate the cyclic adenosine monophosphate (cAMP)-dependent PKA and cyclic guanosine monophosphate (cGMP)-dependent protein kinase G (PKG), which contribute to hyperalgesia (NO-cGMP-PKG) [25]. This activation induces the signaling mechanism through second messengers such as mitogen-activated protein kinases (MAPK) and the nuclear factors kappa B (NF-κB) [25]. This pathway activation was reported in diabetic neuropathy, paclitaxel-induced peripheral neuropathy, and nerve injury [26,27,28]. Therefore, a pharmacological approach that induces TRPV4 downregulation could decrease neuropathic pain due to a reduction in NO production [25]. Then, TRPV4 activation may be mediated by direct agonist production or be sensitized by diverse mechanisms.

Recently, it has been described that the GSK2798745, a new TRPV4 antagonist, has been well-tolerated in healthy volunteers in Phase I clinical trials [22]. Therefore, this compound administration did not induce adverse effects, suggesting that TRPV4 antagonists could be a potential therapeutic target in pain conditions, such as inflammatory pain, neuropathic pain, cancer pain, and migraines. The TRPV4 expressions in the DRG, peripheral fibers, and spinal cord of healthy patients were observed by immunohistochemistry. However, in diabetic patients with neuropathy, the TRPV4 expression did not change in the skin nerve fibers. The non-increased TRPV4 expression suggests the agonist effect on constitutive receptors possibly generated neuropathic pain. However, this is the only article published with human subjects, so there is a need for future studies on humans [29].

Furthermore, the TRPV4 knockout (*Trpv4^−^*^/*−*^) mice did not show impaired heat or touch sensation. However, in inflammatory pain, mechanical allodynia, hyposmotic solution-induced nociception, edema formation and cytokine release were reduced in these animals [30]. In addition, a fly model of neuropathy observed that mutations within the TRPV4 cause disruption of axonal interactions and dendritic degeneration [31]. The TRPV4 role has been researched in models that induce neuropathic pain, such as trauma, surgery, chemotherapy, diabetes and alcohol intake [32,33,34,35,36]. Subsequently, this review will summarize the different rodent models that evaluated TRPV4 involvement in neuropathic pain.

## 3. Neuropathic Pain Induced by Trauma and Surgery and TRPV4 Role

Peripheral neuropathic pain can be caused by trauma and surgery in postoperative patients [37]. Therefore, different animal models were developed to elucidate distinct sorts of injury, such as chronic compression of the dorsal root ganglion (CCD), spinal cord injury (SCI), and infraorbital nerve injury (IONI) [38,39,40,41]. The symptoms detected in animals were similar to those observed in patients with neuropathic pain, such as spontaneous pain, mechanical allodynia, and thermal hyperalgesia [42].

The CCD is a neuropathic pain model that represents the lesions observed in patients with radicular pain, which can be caused by nerve root compression and DRG by herniated intervertebral discs and spinal canal stenosis [38,43,44]. This model is carried out by stainless-steel rods’ unilateral implantation after exposing the L4/L5 intervertebral foramen (one rod for each vertebra), causing a chronic compression of the lumbar DRG [45]. After the CCD procedure, animals develop spontaneous nociception, unilateral mechanical allodynia, and thermal hyperalgesia. The nociception detected in this model is accompanied by enhanced neuronal excitability of compressed DRG [38,43]. Thus, different research groups studied the role of the TRPV4 activation in CCD-induced nociception and increased neuronal excitability.

Initially, using the CCD model in rats, an increase was detected at 7–28 days post-CCD in the levels of TRPV4 mRNA and protein expression in the DRG (using real-time RT-PCR and Western blotting–WB analysis), with the highest level recorded at day 7 post-CCD. Additionally, the treatment with intrathecal (i.t.) administration of TRPV4 antisense (AS) oligodeoxynucleotide (ODN), but not mismatch (MM) ODN, partially reversed the mechanical allodynia and reduced the TRPV4 protein expression in the DRG samples when compared to the control (TRPV4 MM ODN) in this model. In addition, in Ca^2+^ imaging, performed 7 days post-surgery, there was an increase in the DRG neurons responsive to hypotonic solution and 4α-PDD (TRPV4 agonists) in CCD animals compared to the control. The TRPV4 AS ODN treatment reduced the increase in Ca^2+^ influx provoked by both TRPV4 agonists in the DRG of CCD-induced rats. However, the injection of TRPV4 AS ODN did not alter the mechanical threshold of the sham animals. Thus, this study showed that the TRPV4 receptor is involved in mechanical allodynia detection, probably by upregulation of TRPV4 in the DRG [46].

The TRPV4 receptor mediates thermal hyperalgesia in mice after inflammation [47]. Therefore, another author tested the hypothesis that the TRPV4-NO-cGMP-PKG cascade could be involved in maintaining thermal hyperalgesia following CCD surgery in rats. Treatment with i.t. ruthenium red (a TRPV4 non-selective antagonist) and TRPV4 AS ODN demonstrated an antinociceptive effect. In addition, these treatments reduced the nitrite (a marker for NO formation) production in the DRG of CCD rats compared to CCD vehicle-treated animals. The suppression of NO synthesis by the i.t. injection of L-NAME (a non-specific NO synthase inhibitor) decreased thermal hyperalgesia. A similar effect was detected for 1H-[1,2,4]-oxadiazolo [4,3-a] quinoxalin-1-one (ODQ, a soluble guanylate cyclase inhibitor) and Rp-8-pCPT-cGMPS (a PKG inhibitor). Additionally, L-NAME administration reduced nitrite production, but 4α-PDD (the TRPV4 agonist) diminished the antinociceptive effects and enhanced the nitrite production when co-administered with this NO synthase inhibitor in the CCD model. Thus, the TRPV4 activation-induced Ca^2+^ influx generates thermal hyperalgesia due to the activation of NO-cGMP-PKG in the CCD model [48].

In addition, a different study assessed whether NF-κB contributed to the TRPV4-NO pathway in CCD-induced thermal hyperalgesia in rats. The i.t. injection of two NF-κB inhibitors (pyrrolidine dithiocarbamate, PDTC, and BAY11-70082) caused antinociceptive action and reduced NO production (detected as total nitrate plus nitrite) in the DRG samples after CCD induction when compared to the control. Moreover, the TRPV4 agonist (4α-PDD, i.t.) administration reduced the antihyperalgesic effects of these NF-κB inhibitors and attenuated the reduction in NO production in this model. The CCD model caused an increase in NF-κB and decreased inhibitory-kappa B (I-κB) expression in the DRG, which were reversed by the i.t. PDTC injection. However, 4α-PDD administration prevented the effects of PDTC in NF-κB and I-κB expression. Then, the TRPV4 activation may contribute to NO synthesis by the NF-κB pathway signaling and generating neuropathic pain [49].

TRPV4 activation increases CCD-induced neuropathic hyperalgesia by the NO cascade, which could be mediated by Ca^2+^ influx caused by this ion channel. Then, Wang et al. (2015) [50] showed that the i.t. injection of lentiviral vector (LV) containing TRPV4 siRNA (LV–TRPV4) restored the TRPV4 expression (mRNA and protein content) in CCD rats, and the control group (lentiviral vector negative control, LV-NC) showed an enhanced level of this receptor in the DRG samples. Moreover, the LV-TRPV4 administration prevented the development of mechanical allodynia and thermal hyperalgesia after CCD surgery. LV-TRPV4 treatment also decreased the NF-κB level and NO content in the DRG of CCD rats compared to the control (LV-NC). Finally, the DRG neurons challenged with 4α-PDD had a higher level of Ca^2+^ influx in CCD rats when compared to the group previously treated with LV–TRPV4. The intraperitoneal (i.p.) injection of mibefradil (a Ca^2+^ channel inhibitor) decreased NF-κB signaling, NO content and neuropathic thermal hyperalgesia in CCD rats. Additionally, the pre-treatment with 4α-PDD reduced the effects of mibefradil. In this view, the TRPV4-mediated Ca^2+^ influx results in NF-κB activation in the DRG generating neuropathic hyperalgesia in the CCD model [50].

Another study investigated the relationship between TRPV4 and the p38 MAPK pathway in nociception detected in the CCD model. CCD-induced rats showed mechanical allodynia that was reduced by ruthenium red or SB203580 (a P38 inhibitor) also, nociception was enhanced by 4α-PDD or the anisomycin (an agonist of p38) i.t. injection. The injection of ruthenium red or SB203580 reduced TRPV4, p38 and phosphorylated p38. However, 4α-PDD or anisomycin injection increased the TRPV4, p38, and phosphorylated p38 protein expression in the DRG of CCD animals. Moreover, the TRPV4 and p38 protein distribution in the DRG neurons increased after CCD, which was reduced by ruthenium red or the SB203580 injection, or were increased by 4α-PDD or anisomycin administration. The electrophysiological study showed an increase in ectopic discharges of the DRG neurons in CCD rats, and this property was modulated by the agonists (4α-PDD) or inhibitors (ruthenium red) similar to the other analysis [51] (Table 1).

Furthermore, two subsequent articles tested the effect of colchicine-induced microtubule depolymerization on TRPV4-induced nociception detected in the CCD model in rats. The colchicine i.t. injection showed a dose-dependent reduction in mechanical and thermal allodynia in CCD rats, and these reductions were related to a significant decrease in TRPV4 mRNA and protein expression. Additionally, colchicine i.t. administration reduced the TRPV4-mediated currents in the DRG neurons of CCD animals, and a similar result was detected for this compound using cultured HEK cells with TRPV4 expression [57]. A subsequent study observed a dose-dependent inhibitory effect of colchicine on 4α-PDD-induced mechanical and heat hyperalgesia in the CCD model. Colchicine also reduced Ca^2+^ influx and substance P release mediated by 4α-PDD in the DRG cells in vitro [58]. Thus, it is possible that microtubule dynamics interfered with the TRPV4 activation, but further studies should be performed to better describe this mechanism.

The next article used GSK1016790A, a TRPV4 agonist, to investigate whether aquaporin 1 (AQP1) is activated by intracellular cGMP and could modulate CCD-induced allodynia via TRPV4 signaling. The reduction in mechanical allodynia caused by AQP1 lentivirus was attenuated, following the injection of GSK1016790A. Moreover, the increased expression of TRPV4 and AQ1 detected in the DRG and the spinal cord was reduced by the AQP1 lentivirus or AQP1 inhibitor, i.t. injection [52]. Only this article explored the relationship between AQP1 and TRPV4, a pathway that should be further investigated.

Nevertheless, different mechanisms are involved in increased neuronal excitability and TRPV4 activation/upregulation in the CCD model. After that, the Ca^2+^ influx causes the activation of the p38 MAPK, NF-κB pathway, NO production, PKC and AQP1 activation, which cause mechanical and cold allodynia (Figure 2). However, the lack of studies that used a selective antagonist, such as HC067047 or *Trpv4^−^*^/*−*^ mice, impaired the investigation of the role of this channel in the CCD model of neuropathic pain (Table 1).

SCI patients presented temporary or permanent lesions on the spinal cord that altered its function. Traumatic SCI is observed after physical impacts, such as motor vehicle injury, fall, sports-related injury or violence. At the same time, non-traumatic SCI occurs after a tumor, infection, or degenerative disease. The SCI pathophysiology includes neurons and glial cell death, ischemia and inflammation related to neuropathic pain development [59]. The SCI model is used to assess central neuropathic pain in the lower thoracic area that is briefly exposed before a T10 laminectomy is performed [41].

In this model, the initial injury is followed by different processes, such as damage to endothelial, neuronal, or glial cells, especially in the epicenter of the injury, accompanied by vascular disturbance and scarring. TRPV4 activation may cause alteration in the blood-cerebrospinal fluid barrier by disintegrating the cell junctions. Thus, Kumar et al. (2020) [53] observed an increase in TRPV4 expression (especially in endothelial cells) during the inflammatory/acute phase of SCI in female rats and mice. TRPV4 expression progressively increased with the injury’s severity at the epicenter. The TRPV4 antagonist treatment attenuated TRPV4 expression and inflammatory markers, whereas the agonist (GSK1016790A) showed contrary effects, enhancing inflammatory markers in the spinal cord after SCI (Table 1). Moreover, TRPV4 deletion protects endothelial cells from damage, reduces spinal inflammation and scarring, and enhances functional recovery. Mice with TRPV4 deletion were also protected from the induction of thermal hyperalgesia after SCI. Thus, this study showed the relevance of TRPV4 expression in predicting the level of SCI, and TRPV4 activation impairs SCI recovery [53]. Only this study explored the role of TRPV4 in SCI, but it brought different approaches and mechanisms and showed that TRPV4 blockage might also be involved in the disease progression, not only pain induction.

Orofacial pain is detected in distinct pain conditions, including headache, temporomandibular joint disorders, trigeminal neuropathy, and dental pain. Trigeminal neuropathic pain causes include maxillary bone fracture, tooth extraction, or dental implant placement [60,61]. This pain can be chronic and is a frequent social and medical challenge. Patients with trigeminal nerve injury presented intractable mechanical or thermal allodynia in the orofacial region. Then, studying the mechanisms that can cause the sensitization of the trigeminal nerve nociceptors could help to promote a better treatment, such as TRP channel antagonism [62]. TRPV4 is expressed in trigeminal ganglion neurons and other cells, including temporomandibular joint fibroblast-like synoviocytes of rats [63,64]. Therefore, the TRPV4 involvement in orofacial neuropathic pain has been investigated in models of IONI (Table 1).

In this view, the model of IONI used the mechanical head-withdrawal threshold to evaluate the orofacial nociception. This model is caused by exposing the infraorbital nerve by a 10 mm incision along the gingiva–buccal margin proximal to the first molar and a tight ligation of one-third of the infraorbital nerve thickness [39]. Moreover, the expression of TRPV4 was increased in the trigeminal ganglion ipsilateral to the nerve injury, which innervated the whisker pad skin. This increase was reduced by an insulin-like growth factor 1 (IGF-1) antagonist (tranilast) injection to the whisker pad. The treatment of the TRPV4 antagonist (HC-067047, subcutaneous, s.c., injection to the whisker pad) blocks the nociceptive behavior. Thus, the activation of TRPV4 and IGF-1 signaling in the trigeminal ganglion neurons that innervated the whisker pad skin is related to the mechanical allodynia [54].

Similarly, Ando and collaborators (2020) [55] evaluated the functional characteristics of TRPV4 in the same IONI model in rats [55]. The head-withdrawal threshold was reversed by the TRPV4 antagonists (HC-067047 and RN1734) injection to the whisker pad in the IONI-induced group. However, the TRPV4 antagonist treatment (RN1734) did not reduce orofacial heat hyperalgesia, which was only decreased by the TRPV1 antagonist (SB366791) injection. The TRPV4 expression in the trigeminal ganglion neurons was also increased 5 days post-induction of IONI. The oxytocin receptor is involved in descending and ascending nociception pathways in the brain by changing the function of the ion channel, such as TRPV4, and generating an antinociceptive effect [65]. Thus, the peripheral injection of oxytocin could impair TRPV4 increased expression and orofacial nociception, and these effects could be partially suppressed by oxytocin receptor antagonism (atosiban, s.c. local administration) [55].

Subsequently, the same model of trigeminal neuralgia was used to evaluate the TRPV4 role in this type of neuropathic pain. After IONI, the nociceptive behavior was increased, and the expression of TRPV4 was also raised in the trigeminal spinal subnucleus caudalis. Therefore, the s.c. injection of botulinum toxin type A (BoNT-A) in the whisker pad reduced TRPV4 expression. BoNT-A inhibits the neurotransmitter release reducing the peripheral sensitization and modulates the TRPV1 expression in primary sensory neurons [66]. Thus, TRPV4 activation can mediate mechanical hyperalgesia in the IONI model, but further experiments should be conducted to explore this mechanism. BoNT-A treats trigeminal neuropathic pain, but the mechanism should be studied since this treatment also reduces TRPM3 expression [56].

Thus, TRPV4 is involved in mechanical allodynia after IONI induction, and there is also an increased content of this receptor after lesion, a factor that could lead to the hypersensitivity observed in this model. However, the mechanism causing TRPV4 increased expression should be better investigated.

## 4. Chemotherapy-Induced Neuropathic Pain (CINP) and TRPV4 Participation

Chemotherapy-induced neuropathic pain (CINP) is a common side effect of different antineoplastic drugs, such as paclitaxel (Taxol^®^), thalidomide (Thalomid^®^) and vincristine (Oncovin^®^) [67]. Among these drugs, paclitaxel is used in solid tumors, while vincristine is used for hematological cancers. Both paclitaxel and vincristine administration induced numbness, pain, and burning sensation in the hands and feet. In some cases, neuropathy can be irreversible, reducing the patient’s quality of life and interfering with daily activities [68]. Like paclitaxel and vincristine, thalidomide is associated with developing peripheral neuropathy [68]. However, the mechanism that underlines CINP is still not fully understood. In this view, pre-clinical models investigate the TRPV4 channel role in CINP (Table 2) [67].

The paclitaxel-induced peripheral neuropathy model was first used to evaluate the TRPV4 involvement in nociceptive behavior responses to mechanical and hypotonic stimulation of the hind paw. TRPV4 AS ODN i.t. treatment reversed the nociceptive behavior compared to the TRPV4 MM ODN in the model of CINP in rats. The injection of hypotonic saline (a TRPV4 activation) caused nociceptive behavior in rats previously treated with paclitaxel, which TRPV4 AS ODN i.t. treatment partially reduced. However, there was no alteration in the TRPV4 expression in the saphenous nerve of paclitaxel-treated animals. In addition, the injection of GRGDTP (an integrin antagonist peptide) in the hind paw reduced mechanical hyperalgesia and hypotonicity-induced nociception in paclitaxel-injected groups. Moreover, an Src family kinase-specific inhibitor was used to evaluate the integrin/Src tyrosine kinase phosphorylation. The kinase-specific inhibitor (PP1, Src family kinase-specific inhibitor and genistein, nonspecific tyrosine kinase inhibitor) treatment in the hind paw reversed the paclitaxel-induced mechanical hyperalgesia and the hypotonicity-induced nociception. These behavioral studies were also accompanied in vitro by Ca^2+^ imaging assays in cultured sensory neurons that showed a Ca^2+^ influx after the paclitaxel treatment. Therefore, TRPV4 contributes to enhanced nociception in the paclitaxel-induced neuropathy model in rats by activating primary afferent nociceptors depending on the integrin/Src kinase pathway [69].

A second study from this author investigates the TRPV4 participation in paclitaxel- and vincristine-induced mechanical hyperalgesia and hypotonicity-induced nociceptive behavior. The TRPV4 AS ODN i.t. treatment reduced vincristine-induced mechanical allodynia and hypotonicity-induced nociception. Moreover, only *Trpv4^+^*^/*+*^ mice developed mechanical hyperalgesia after the chemotherapy administration (paclitaxel and vincristine), and *Trpv4^−^*^/*−*^ mice were protected from the induction of nociception in these models of CIPN. Furthermore, the i.t. injection of α_2_ integrin AS ODN reduced paclitaxel-induced mechanical allodynia and hypotonicity-induced nociception in rats. Src tyrosine kinase and TRPV4 interaction may lead to integrin activation, as activation of the Src tyrosine kinase by YEEIP (Src tyrosine kinase activator peptide) caused TRPV4-dependent mechanical hyperalgesia. Thus, in nociceptors, TRPV4 is involved in mechanical hypersensitivity after paclitaxel and vincristine injection, probably by a pathway involving α_2_β_2_ integrin and Src tyrosine kinase in sensory neurons [70].

Moreover, 20,30-dideoxycytidine (ddC) is used in HIV therapy generating painful peripheral neuropathy [76]. However, since the specifics of ddC-induced neuropathy mechanism remains unclear, rodent models are used to investigate it. In rats, the model is induced by ddC injection in the tail vein (50 mg/kg, intravenously i.v., single dose) [77]. After induction, hypotonicity-induced nociceptive behavior was observed in ddC-treated rats, and the treatment with TRPV4 AS ODN reversed it [70]. However, this was the only study that evaluated the TRPV4 role in ddC-induced peripheral neuropathy but introduced the perspective that the TRPV4 blockage could be a treatment approach [70].

Subsequently, the same research group hypothesized the TRPC6/TRPC1 and TRPV4 involvement in mechanical hyperalgesia in the paclitaxel-induced model. TRPC1 and TRPC6 are GsMTx-4-sensitive SACs (stretch-activated ion channels). They showed that TRPV4, TRPC1, and TRPC6 were co-expressed in the DRG neurons. GsMTx-4 (a TRPC1 and TRPC6 inhibitor) intraplantar injection (i.pl.) reduced paclitaxel-induced mechanical hyperalgesia and hypotonicity-induced nociception. Consequently, these channels contribute to the TRPV4 detection of mechanical nociceptive stimuli in sensitized primary afferent nociceptors in the paclitaxel-induced neuropathic pain model [71].

Another study evaluated if PAR2 activation caused sensitization of TRPV4, TRPV1 and TRPA1 in a paclitaxel-induced CINP model in mice. PAR2 is co-expressed with TRPV4, TRPV1, and TRPA1 in sensory neurons and can be activated by tryptase. PAR2 receptors can cause the activation of PKA and PKC (throughout phospholipase C activation, PLC), generating the sensitization of ionic channels, such as, TRPA1, TRPV1 and TRPV4. The nociceptive behavior was measured by mechanical/cold allodynia and heat hyperalgesia. PAR2 antagonist (FSLLRY-amide) i.t. injection reversed paclitaxel-induced nociception. The treatment with a TRPV4 antagonist (RN1734, i.p.) partially reversed the mechanical allodynia and heat hyperalgesia but did not alter cold allodynia. Moreover, these authors used PKA (KT5720), PKC (myristyolated EAVSLKPT), and PLC (U73122) inhibitors that PAR2 activation by tryptase may cause TRPV4 sensitization via PKA/PKC activation. The TRPV1 and TRPA1 could be involved in this type of neuropathy [72].

Furthermore, another research group showed that the administration of TRPV4 antagonist (HC-067047) partially reduced mechanical allodynia caused by a paclitaxel single injection in mice. However, the co-injection of a TRPV4 antagonist (HC-067047, i.p.) and a TRPA1 antagonist (HC-030031, intragastric (i.g.)) completely reduced the mechanical hypersensitivity. However, cold allodynia is solely mediated by TRPA1 activation. Paclitaxel could not directly activate the TRPV4 or TRPA1 in the DRG Ca^2+^ influx assay. However, this chemotherapy caused the release of calcitonin gene-related peptide (CGRP) from slices of mouse esophagus, which was reduced by the combination of TRPV4 and TRPA1 antagonists and by an antioxidant compound (glutathione). Thus, the paclitaxel injection mediates the induction of mechanical allodynia by TRPA1 and TRPV4 activation and reactive compound production [73].

Moreover, paclitaxel can interact with Toll-like receptor 4 (TLR-4) and then mediate the production of tumor necrosis factor-α (TNF-α). Thus, subsequently, a paclitaxel-induced peripheral neuropathy model was used to investigate the TRPV4 expression in the DRG neurons and its relation to nociceptive behavior due to the TLR-4 activation and the production of TNF-α. Paclitaxel increases TNF-α, TRPV4 and TRPA1 expression in the DRG samples in mice. Moreover, the i.pl. treatment with the replication-defective herpes simplex virus (HSV)-based vector blocked the TNF-α signaling, and consequently, it inhibited mRNA expression of TRPV4 and TRPA1 and nociceptive behavior. Additionally, in vitro, paclitaxel increased TNF-α release and expression in enriched primary satellite cells, which was blocked by a TLR4 antagonist (TAK-242). Then, recombinant TNF-α application increased the expression of TRPV4 and TRPA1 in cultured DRG neurons in vitro. Hence, paclitaxel binding to TLR-4 in satellite glial cells induces the production of TNF-α that enhances the expression of TRPA1 and TRPV4 in the DRG neurons to cause mechanical and cold allodynia [74].

A different study investigated the participation of kinin receptors in TRPV4 sensitization via PKCε activation, a mechanism that could be involved in paclitaxel-induced mechanical hyperalgesia. The pro-algesic effects of kinins can be mediated by sensitization of TRPV1 and TRPA1 channels. Since TRPV4 also mediates paclitaxel mechanical hypersensitivity, this channel might be implicated in the nociceptive actions of kinins. First, the authors demonstrated that kinin agonists (bradykinin and des-Arg9-bradykinin, i.pl. injection) induced mechanical hypersensitivity, which was partially reduced by TRPV4 antagonist (HC-067047, i.p.) treatment. Moreover, the HC-067047 injection inhibited paclitaxel-induced mechanical hyperalgesia and hypotonic solution-induced nociception (i.pl.), and a similar effect was observed for kinin B_1_R (DALBK) and B_2_R (HOE 140). Paclitaxel increased the expression of PKCε, and the kinins receptor antagonists, i.p. injection, prevented this enhancement of PKCε expression in the plantar skin and in the DRG. Therefore, kinins sensitize TRPV4 and cause mechanical hyperalgesia in paclitaxel-induced peripheral neuropathy through PKCε activation [23].

The role of TRPV4 and TRPA1 activation was also evaluated in a model of thalidomide, and its derivatives (pomalidomide and lenalidomide) evoked mechanical and cold allodynia. The TRPV4 or TRPA1 antagonist (i.p.) partially attenuated mechanical allodynia, and a combination of TRPA1 and TRPV4 antagonists completely reversed thalidomide-, pomalidomide- and lenalidomide-evoked mechanical allodynia. In contrast, the cold allodynia was exclusively dependent on TRPA1 activation. These results were confirmed with mice with genetic deletion since thalidomide and its derivatives evoked mechanical allodynia that was partially reduced in *Trpa1^−^*^/*−*^ and *Trpv4^−^*^/*−*^ mice. Thalidomide, pomalidomide, and lenalidomide did not cause a Ca^2+^ influx in the DRG neurons in vitro, but hydrogen peroxide activated these channels. Additionally, α-phenyl-tert-butyl-nitrone (PBN, a ROS scavenger) injection in mice reduced mechanical and cold allodynia caused by a thalidomide injection and decreased the levels of hydrogen peroxide and 4-hydroxynonenal in tissues (lumbar spinal cord, plantar skin, and sciatic nerve) after thalidomide administration. Moreover, by injecting the TRPA1 and TRPV4 antagonists using the i.t. or i.pl. route, the authors showed that spinal TRPV4 and plantar TRPA1 were relevant to the induction of mechanical allodynia after thalidomide injection. This article showed similar results to the model of CIPN caused by paclitaxel injection, where TRPA1 and TRPV4 were responsible for mechanical allodynia induction. Additionally, hydrogen peroxide was described as a TRPV4 agonist, with less potency to activate this cation channel when compared to TRPA1 [77].

Most articles that studied CIPN-induced nociception used the paclitaxel injection model (Figure 3), and all of the studies used male rodents. The role of TRPV4 was investigated in mechanical and hypotonic-induced nociception. As described above, distinct mechanisms could sensitize the TRPV4 (α_2_β_2_ integrin, Src tyrosine kinase, PAR2, kinins, and PKC). Moreover, two articles proposed the activation of this channel by reactive compounds (and showed the TRPV4 direct activation by hydrogen peroxide). One article proposed the direct activation of TRL4 by paclitaxel in satellite cells in the DRG and the release of TNF-α to induce TRPV4 increased expression in the DRGs. However, the chemotherapeutics studied do not seem directly related to TRPV4 activation. Additionally, the TRPV4 increased expression was not described in all studies or was not evaluated as the cause of nociception.

## 5. Other Neuropathic Pain Models with TRPV4 Involvement

Moreover, another neuropathic pain rodent model that evaluated the TRPV4 channel role is painful diabetic neuropathy. The physiopathology of diabetic neuropathy involves hyperglycemia, oxidative stress, and mitochondrial dysfunction, which could generate TRPV4 activation. Moreover, investigating the TRP role in painful diabetic neuropathy could represent a new treatment approach since the current pain therapies are insufficient. Three studies assessed the TRPV4 channel in painful diabetic neuropathy rodent models using different pharmacological approaches [34,70,78].

Painful diabetic neuropathy can be caused by a streptozotocin injection in rodents, and the blood glucose was evaluated to confirm hyperglycemia induction by this model. The streptozotocin model in rats caused mechanical hyperalgesia and hypotonicity-induced nociceptive behavior. The TRPV4 AS ODN i.t. treatment reduced both streptozotocin-induced mechanical allodynia and hypotonicity-induced nociception in rats. Only *Trpv4^+^*^/*+*^ mice developed mechanical hyperalgesia after the diabetes induction [70].

Subsequently, Dias and collaborators (2019) [34] investigated TRPV4 expression and its association with mechanical and cold allodynia in streptozotocin-induced diabetes in mice. A single dose of TRPV4 antagonist (HC-067047, s.c.) reduced mechanical allodynia. Additionally, the repeated administration of HC-067047 (s.c.) for 6 weeks prevented the mechanical allodynia induction, but it did not alter cold allodynia or hyperglycemia. The repeated injection of this TRPV4 antagonist did not change the mice’s weight, temperature or locomotor activity. However, the immunohistochemical expression of TRPV4 channel expression did not change after 6 weeks post-induction in the sciatic nerve, the DRG, or hind paw plantar skin, suggesting the antagonist had an effect on the constitutive receptors. However, the authors did not evaluate the TRPV4 expression using a quantitative method [34].

Moreover, a different research group showed that TRPV4 expression was increased 2 weeks post-induction of the painful diabetic neuropathy model in rats. However, it decreased after 3 weeks in the DRG and spinal dorsal horn. Moreover, the rat showed mechanical allodynia, and the TRPV4 antagonist (HC-067047, i.t.) single or repeated treatment reduced nociception. The single i.t. injection of a TRPV4 agonist GSK1016790A enhanced the mechanical allodynia after 2 weeks of model induction [78]. These authors also published a distinct paper showing a similar profile of increased expression for TRPV1 in the DRG and spinal cord and the antinociceptive effect of i.t. injection of TRPV1 antagonists for mechanical allodynia in the painful diabetic neuropathy model in rats [79]. However, mechanical allodynia, as described above [34], can be seen even after 2 weeks of streptozotocin injection. Thus, TRPV4 should be activated by an endogenous agonist or sensitized by distinct mechanisms to be studied in further detail.

Furthermore, alcohol-induced peripheral neuropathy is a long-time effect of alcohol consumption characterized by pain in the extremities described as spontaneous burning pain, hyperalgesia, and allodynia [80]. Among the mechanisms evaluated, alcohol consumption raises oxidative–nitrosative stress and causes pro-inflammatory cytokine release associated with PKC activation [81]. In this model, rats were fed with 6.5% ethanol for four days and traditional food for the last three days of the week. Mechanical thresholds decreased in alcohol-treated rats and they showed hypotonicity-induced nociceptive behavior. However, the level of TRPV4 protein expression in alcohol-fed rats did not change in the saphenous nerve 2 weeks after alcohol exposure. The treatment with TRPV4 AS ODN reversed both nociceptive behaviors. In addition, the authors investigated the contribution of α_2_ integrin and Src tyrosine kinase in mechanical hyperalgesia with an α_2_ integrin AS ODN and Src family tyrosine kinase inhibitor. These treatments reversed the alcohol-induced mechanical hyperalgesia and hypotonicity-induced nociceptive behavior, but this result did not directly show the interaction between Src tyrosine kinase and TRPV4. Then, YEEIP induced TRPV4-dependent mechanical hyperalgesia, and the ODN treatment reversed this behavior. Additionally, the direct interaction of TRPV4 with α_2_ integrin and Src tyrosine kinase was demonstrated by co-immunoprecipitation using DRG neurons isolated from alcohol-fed rats. Therefore, α_2_β_1_ integrin and TRPV4 interaction transduces mechanical allodynia in the injured sensory neurons of the alcohol-induced peripheral neuropathy model [70]. Only this study investigated alcohol-induced neuropathic pain in rats and TRPV4 mice with genetic deletion (Table 3). It showed a similar mechanism of paclitaxel-induced nociception as described above in this review.

## 6. Conclusions

The participation of TRPV4 in different models of neuropathic pain was evaluated using pharmacological and genetic approaches. All the studies detected the antinociceptive effect of TRPV4 antagonists and the reduction in nociception after TRPV4 expression reduction (AS ODN i.t. injection) or in *Trpv4^−^*^/*−*^ mice. The activation of TRPV4 may be mediated by reactive compounds (hydrogen peroxide), or different molecular pathways could sensitize this channel. In the models of neuropathic pain caused by trauma or surgery, the upregulation of spinal cord TRPV4 was described in most of the studies. No adverse effects were related to TRPV4 antagonists or genetic strategies of TRPV4 ablation, and TRPV4 is not involved in touch or thermal perception in naive mice.

Novel studies should also investigate the role of TRPV4 in spontaneous pain detected in these models of chronic pain and evaluate the antinociceptive effect of TRPV4 antagonists on female animals. Moreover, only hydrogen peroxide and NO were evaluated as TRPV4 agonists, so one possible target to reduce neuropathic pain should focus on decreasing these oxidated compounds. The expression of TRPV4 in the central nervous system should be further investigated since only one study used a model of central neuropathic pain (SCI model). Then, TRPV4 could be a promising pharmacological strategy to control pain observed in neuropathic pain models, especially mechanical allodynia and thermal hyperalgesia.

## Figures and Tables

**Figure 1 antioxidants-12-00024-f001:**
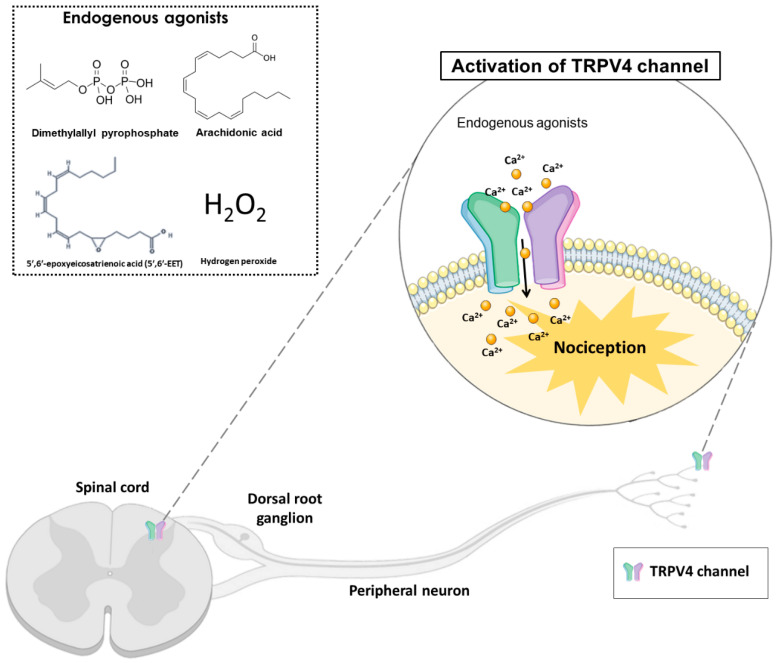
The activation of transient receptor potential vanilloid 4 (TRPV4) by endogenous agonists (dimethylallyl pyrophosphate, arachidonic acid, 5′,6′-epoxyeicosatrienoic acid, and hydrogen peroxide) in the dorsal spinal cord and peripheral nociceptive neurons, may cause nociception in different pain models. TRPV4 is a non-selective calcium channel that mediates calcium influx, promoting nociception transduction.

**Figure 2 antioxidants-12-00024-f002:**
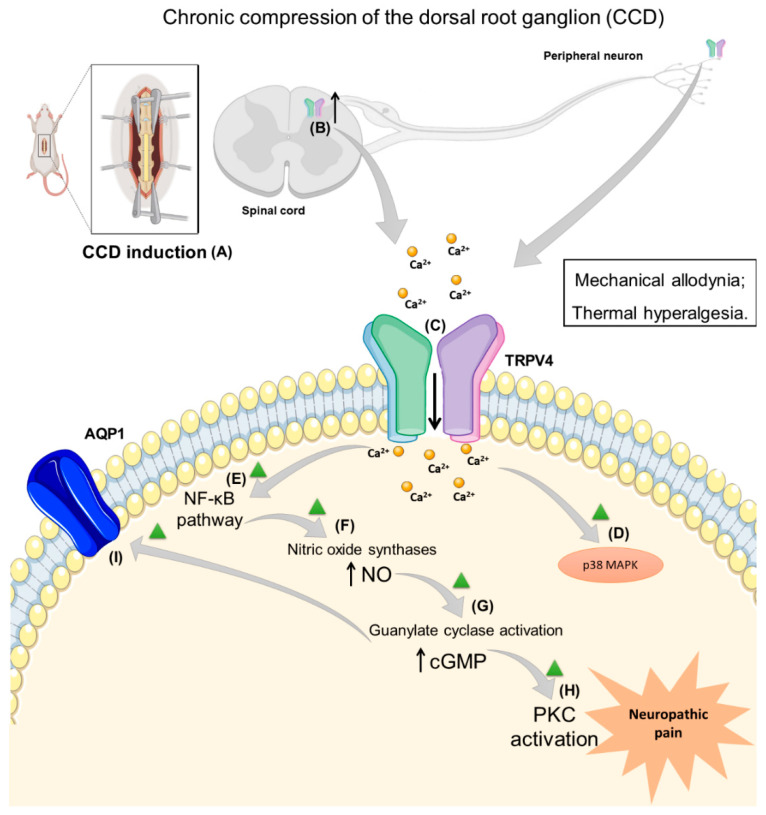
(**A**) Neuropathic pain induction due to transient receptor potential vanilloid 4 (TRPV4) activation in the spinal dorsal horn in a model of chronic compression of the dorsal root ganglion (CCD). (**B**) The expression of the TRPV4 receptor increased in the spinal dorsal horn, and dorsal root ganglion after CCD induction; (**C**–**E**) Calcium influx throughout the TRPV4 channel causes the activation of p38 mitogen-activated protein kinases (MAPK) and nuclear factor-kappa (NF-κB) pathway. (**F**) NF-κB causes the activation of nitric oxide synthase leading to nitric oxide (NO) production. (**G**) NO-induced cyclic guanosine monophosphate (cGMP) production by guanylate cyclase. (**H**,**I**) Increased cGMP production causes the activation of protein kinase C (PKC) and aquaporin 1 (AQP1).

**Figure 3 antioxidants-12-00024-f003:**
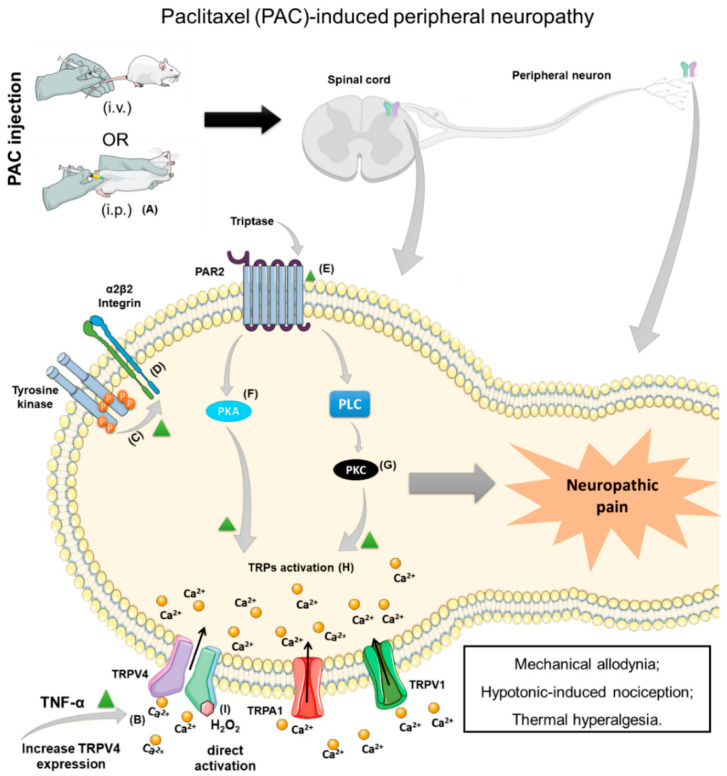
(**A**) Paclitaxel (PAC)-induced peripheral neuropathy due to transient receptor potential vanilloid 4 (TRPV4) activation in the dorsal spinal cord and peripheral nociceptive neurons in a model of paclitaxel (PAC)-induced CINP by intravenous (i.v.) or intraperitoneal (i.p.) injection. (**B**) TNF-α increases the TRPV4 expression in the DRG neurons. (**C**,**D**) Tyrosine kinase receptor phosphorylation may lead to α_2_β_2_ integrin activation. (**E**–**G**) The activation of protease activating receptor 2 (PAR2) by tryptase induces protein kinase C (PKC) and protein kinase A (PKA) activation that causes (**H**) TRPA1, TRPV1, and TRPV4 gating. (**I**) TRPV4 direct activation by hydrogen peroxide (H_2_O_2_) causes calcium influx that induces neuropathic pain.

**Table 1 antioxidants-12-00024-t001:** Rodent trauma and surgery models that induce neuropathic pain due to TRPV4 activation.

Rodent Type	Model	Tissue	TRPV4 Expression Verification Method	Treatment	Dose, Time of Treatment, and Route of Administration	References
Rats (non-specific)	CCD	DRG	–	ODN	40 µg i.t. once a day for 7 days	Zhang (2008) [46]
Wistar rats (male)	CCD	DRG	WB	ODN/antagonist (ruthenium red)/agonist (4α-PDD)	40 µg i.t. once a day for 7 days/0.01–1 nmol i.t./1 nmol i.t. single administration	Ding (2010) [48]
Wistar rats (male)	CCD	DRG	–	Agonist (4α-PDD)	1 nM i.t. single administration	Wang (2011) [49]
Wistar rats (male)	CCD	DRG	WB, RT-PCR, and IHC	–	–	Wei (2013) [52]
Wistar rats (male)	CCD	–	–	Agonist (4α-PDD)	5 and 50 nmol/kg i.t. single administration	Ning (2014) [53]
Wistar rats (male)	CCD	DRG	WB, RT-PCR, and IHC	TRPV4 targeted siRNA (LV–TRPV4)	10 µL/day for 3 days (continuously infused by i.t. catheter)	Wang (2015) [50]
Wistar rats (male)	CCD	DRG	WB	Antagonist (ruthenium red)/agonist (4α-PDD)	1–100 nmol/l i.t. single administration	Qu (2016) [51]
Wistar rats (male)	CCD	DRG, and spinal cord	WB and RT-PCR	Agonist (GSK1016790A)	0.5 µM i.t. single administration	Wei (2020) [52]
SD rats, C57BL/6 KO, and WT mice (female)	SCI	Spinal cord	WB, RT-PCR, and IHC	Antagonist (RN1734)/agonist (GSK1016790A)	5 mg/kg, i.p./50 pmol i.t. single administration	Kumar (2020) [53]
SD rats (male)	IONI	TG	IHC	Antagonist (HC-067047)	100 µg/kg s.c. single administration to the whisker pad	Sugawara (2019) [54]
SD rats (male)	IONI	TG	IHC	Antagonists (HC-067047 and RN1734)	HC067047 (30 mg/mL); RN1734 (0.4 g/mL) s.c. single administration to the whisker pad	Ando (2020) [55]
SD rats (male)	IONI	Vc	WB and IHC	–	–	Zhang (2019) [56]

4α-phorbol 12,13-didecanoate (4α-PDD); chronic compression of the dorsal root ganglion (CCD); dorsal root ganglion (DRG); immunohistochemistry (IHC); infraorbital nerve injury (IONI); intraperitoneal (i.p.); intrathecal (i.t.); knockout (KO); oligodeoxynucleotide (ODN); spinal cord injury (SCI); Sprague Dawley (SD); transient receptor potential vanilloid 4 (TRPV4); trigeminal ganglion (TG); trigeminal subnucleus caudalis (Vc); real-time RT-PCR; subcutaneous (s.c.); Western blot (WB); wild type (WT).

**Table 2 antioxidants-12-00024-t002:** Rodent models of chemotherapy-induced neuropathic pain with TRPV4 involvement.

Rodent Type	Model	Tissue	TRPV4 Expression Verification Method	Treatment	Dose, Time of Treatment and Route of Administration	References
SD rats (male)	Paclitaxel 1 mg/mL i.p. once a day for 10 days.	DRG	WB	ODN	40 µg i.t. once a day for 3 days	Alessandri-Haber (2004) [69]
SD rats (male), KO and WT C57BL/6 mice (male)	Vincristine 200 µg/kg i.v. for 5 days (rats) or i.p. single injection (mice). Paclitaxel 1 mg/mL i.p. once a day for 10 days (rats) or 6 mg/kg single i.p. injection (mice).	–	–	ODN	40 µg i.t. once a day for 3 days	Alessandri-Haber (2008) [70]
SD rats (male)	ddC 50 mg/kg i.v. single dose.	–	–	ODN	40 µg i.t. once a day for 3 days	Alessandri-Haber (2008) [70]
SD rats (male)	Paclitaxel 1 mg/kg i.p. once a day for 10 days.	–	–	–	–	Alessandri-Haber (2009) [71]
ICR mice (male)	Paclitaxel 1 mg/kg i.p. every 2 days for 6 days.	–	–	Antagonist (RN1734)	30 mg/kg i.p. single administration	Chen (2011) [72]
C57BL/6 WT and KO mice (male)	Paclitaxel 6 mg/kg i.p. single administration.	–	–	Antagonist (HC-067047)	10 mg/kg i.p. single administration	Materazzi (2012) [73]
SD rats (male)	Paclitaxel 16 mg/kg i.p. once a week for 5 weeks.	DRG	WB, RT-qPCR, and IHC	–	–	Wu (2015) [74]
Swiss mice (male)	Paclitaxel 2 mg/kg i.p. for 5 days.	–	–	Antagonist (HC-067047)	10 mg/kg, i.p.	Costa (2018) [23]
SD rats (male), C57BL/6J, WT and KO mice (male)	Thalidomide 1, 10, 50, and 100 mg/kg, pomalidomide 1 mg/kg, and lenalidomide 5 mg/kg, i.p. single injection.	spinal cord	–	Antagonist (HC-067047)	10 mg/kg i.p. and 100 µg i.t. and i.pl.	De Logu (2020) [75]

Dorsal root ganglion (DRG); immunohistochemistry (IHC); intraperitoneal (i.p.); intraplantar (i.pl.); intrathecal (i.t.); intravenously (i.v.); knockout (KO); oligodeoxynucleotide (ODN); transient receptor potential vanilloid 4 (TRPV4); Sprague Dawley (SD); real-time RT-PCR; Western blot (WB); wild type (WT).

**Table 3 antioxidants-12-00024-t003:** Other neuropathic pain models evaluated TRPV4 activation.

Rodent Type	Model	Tissue	TRPV4 Expression Verification Method	Treatment	Dose, Time of Treatment, and Route of Administration	References
SD rats, KO and WT C57BL/6 mice (male).	Streptozotocin 50 mg/mL i.v. (rat) or 75 mg/kg i.p. (mice).	–	–	ODN	40 µg i.t. once a day for 3 days.	Alessandri-Haber (2008) [70]
SD rats, KO and WT C57BL/6 mice (male).	Ethanol 6.5% i.g. for 4 days.	Saphenous nerve	WB	ODN	40 µg i.t. once a day for 3 days.	Alessandri-Haber (2008) [70]
Swiss mice (male).	Streptozotocin 120 mg/kg i.p. for 2 consecutive days.	Sciatic nerve, DRG, and hind paw plantar skin	IHC	Antagonist (HC-067047)	10 mg/kg, s.c. once a day for 6 consecutive days or 1 mg/kg once a day for 4 weeks.	Dias (2019) [34]
SD rats (male).	Streptozotocin 60 mg/kg i.p. single dose.	DRG, and spinal dorsal horn	WB, RT-PCR, and IHC	Antagonist (HC-067047), and agonist (GSK1016790A)	400 ng/kg i.t. repeated injection (once a day for 7 consecutive days), or 200 ng/kg i.t. single injection.	Cui (2020) [78]

Dorsal root ganglion (DRG); immunohistochemistry (IHC); intragastric (i.g.); intraperitoneal (i.p.); intrathecal (i.t.); intravenously (i.v.); knockout (KO); oligodeoxynucleotide (ODN); transient receptor potential vanilloid 4 (TRPV4); real-time RT-PCR; Sprague Dawley (SD); subcutaneous (s.c.); Western blot (WB); wild type (WT).

## Data Availability

Not applicable.

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
