# Peer review of "TRPV4 Role in Neuropathic Pain Mechanisms in Rodents"

_antioxidants, 2022, doi:10.3390/antiox12010024_

Round 1

Reviewer 1 Report

The review is a well written article concerning the role of the TRPV4 receptor in neuropathy. The topic has been examined in all its complexity and from all points of view and the result is that of an article that fills in a gap in the knowledge of the possible and important roles of this receptor in the treatment and/or prevention of the neuropathic pain, either spontaneous or chemically induced by treatment with anti-tumor drugs.

I do not have substantive observations with the exception of page 5, line 207-208, where the sentence 'this receptor is only involved in the mechanical allodynia detection, probably by upregulation of TRPV4 in DRG' should be rephrased, since a few line above (lines 199-200) is stated that TRPV4 AS ODN reduced the TRPV1 protein expression.

Minor:

the sentences in line 163 'The TRPV4 expression (immunohistochemistry, IHC)' and line 537 'The TRPV4 receptor expression (IHC)' should be revised. The acronym IHC is in fact explained in the legend of Tables and it is misleading. as well as wrong, its definition after the 'TRPV4 expression'. I suggest to delete (immunohistochemistry, IHC) in line 163 and (IHC) in line 537, and simply substitute with 'the immunohistochemical expression of..'.

line 122: nervus --> nervous

Author Response

Manuscript antioxidants-1994060

ANSWERS TO THE COMMENT OF REVIEWER 2

We would like to thank you very much for the important and constructive suggestions for our manuscript. They will certainly improve its scientific value.

Question 1: page 5, line 207-208, where the sentence 'this receptor is only involved in the mechanical allodynia detection, probably by upregulation of TRPV4 in DRG' should be rephrased, since a few line above (lines 199-200) is stated that TRPV4 AS ODN reduced the TRPV1 protein expression.

Answer: As requested, we rephrased the sentence (lines 308-310, page 15).

Question 2: the sentences in line 163 'The TRPV4 expression (immunohistochemistry, IHC)' and line 537 'The TRPV4 receptor expression (IHC)' should be revised. The acronym IHC is in fact explained in the legend of Tables and it is misleading. as well as wrong, its definition after the 'TRPV4 expression'. I suggest to delete (immunohistochemistry, IHC) in line 163 and (IHC) in line 537, and simply substitute with 'the immunohistochemical expression of..'.

Answer: As recommended, we exclude the abbreviation from the text and change the sentences (lines 260-261 and 692, page 13 and 35)

Question 3: line 122: nervus --> nervous

Answer: As requested, we changed the text (lines 207, page 10)

Sincerely yours,

Professor Gabriela Trevisan

Reviewer 2 Report

This review article written by Rodrigues et al. mentions the role of TRPV4 channels located in the DRG, spinal dorsal horn, TG and trigeminal spinal subnucleus caudalis in rodent neuropathic pain models.  Figures 1-3 give schematic diagrams showing cellular mechanisms for neuropathic pain produced by TRPV4 activation, and Tables 1-3 present rodent models of neuropathic pain produced by trauma, surgery, chemotherapy and so on.  There are so many points that should be addressed and may serve to amend this manuscript, as follows:

Major points:

1.     This review article is a little difficult to read.  In order to aid in understanding this article, the authors should provide an abbreviation list of the words used and a list of the drug actions mentioned in this manuscript.

2.     Lines 51 and 52: it should be stated that opioid receptor activation is also involved in tramadol-induced antinociception (see DOI:10.1371/journal.pone.0125147).

3.     Line 87: it seems to be unlikely that “the six family members” indicate TRPC, TRPV, TRPM, TRPP, TRPML and TRPA.  Therefore, this sentence should be revised.

4.     The contents of Figs. 1-3 partially overlap.  It may be better to omit or reduce some of the images shown in Fig. 1 and provide a diagram showing the chemical structure of TRPV4 channels mentioned in the second paragraph on page 3.

5.     Figure 1: correspondence between large circle and small circle is not clear.  Small circle does not seem to be located in the DRG.  Judging from the colors used, TRPV4 seems to be expressed in spinal cord, DRG neuron fiber and peripheral neuron fiber.  Is this OK?  Please reply to this question.

6.     Figure 2: (a), (b), ... in its legend do not correspond to (A), (B), … in Fig. 2.  Please amend this point.  What is the figure shown in the upper left of Fig. 2?  Please reply to this question.

7.     Figure 3: (a), (b), ... in its legend do not correspond to (A), (B), … in Fig. 3.  Please amend this point.  In its legend, please explain shortly the figure (injection method) shown in the upper left of Fig. 3.

8.     Tables 1-3: there should be more space between each item.  Using “SD” instead of “Sprague-Dawley” might save space in each Table.  “Author” should be “Reference”.  It is not necessary to define “TRPV4” twice (title and footnote).  Please amend this point.

Specific points:

1.     Lines 8 and 9: “role” has been written twice.  Please amend this point.

2.     Line 13: “(NO)” should be “(NO),”.

3.     Line 22: not “TRPV1” but “TRPV4”?  Please check about this matter.

4.     Line 68: “are” should be “is”.

5.     Line 87: it may be better for “TRPV1/4” to be changed to “TRPV1-4”.

6.     Line 125: please expand “UVB”.

7.     Lines 163 and 164: “TRPV4 .. IHC)” should be revised in English.

8.     Lines 172 and 173: this sentence should be revised.

9.     Line 204: “agonists” should be “agonist”.

10.  Line 212: please move “i.t” to the correct position.

11.  Line 240: please explain “LV-NC” shortly.

12.  Line 261: please write concrete drugs of “the agonists or inhibitors”.

13.  Line 267: “DRG” should be “DRG neurons”.

14.  Line 283: what kind of “a selective antagonist”?  Please make this point clear.

15.  Line 300: what kind of “an agonist”?  Please make this point clear.

16.  Line 304: not “Thus” but “Thus,“.

17.  The definition of CCD is inconsistent between line 181 and the footnote of Table 1.  Please amend this point.

18.  Last line on page 8, line 476 and the footnote of Table 3: not “real-time” but “real-time PCR”.

19.  Lines 329 and 330: is “a better treatment” “TRP channels”?  Please amend this point.

20.  Line 344: please correct English in this line.

21.  Line 350: please correct the position of “(SB366791)”.

22.  Lines 351-353: it may be better to shortly mention possible cellular mechanisms for this oxytocin action.

23.  Lines 357-361: it may be better to shortly mention possible cellular mechanisms for this “botulinum toxin type A” action.  In general, the abbreviation for “botulinum toxin type A” uses “BoNT-A” but not “BTX-A”.

24.  Line 430: is “PLC inhibitor” “PKA and PKC”?  Some clarification is necessary here.

25.  Line 436: please explain here “i.g.”, although this is given in the footnote of Table 3.

26.  Line 439: please give an expansion of “CGRP”.

27.  Line 512: what is “de TRPV4 expression”?  Please reply to this question.

28.  Line 516: not “causing” but “causes”?  Please reply to this question.

29.  Line 593: please correct English in this line.

30.  There appear to be much more mistakes than pointed out above.  Please check the manuscript very carefully.

Author Response

Manuscript antioxidants-1994060

ANSWERS TO THE COMMENT OF REVIEWER 2

We would like to thank you very much for the important and constructive suggestions for our manuscript. They will certainly improve its scientific value.

Question 1: This review article is a little difficult to read. In order to aid in understanding this article, the authors should provide an abbreviation list of the words used and a list of the drug actions mentioned in this manuscript.

Answer: We appreciate your recommendation, and as a request, we insert the abbreviation list in the manuscript (pages 3 to 5).

Question 2: Lines 51 and 52: it should be stated that opioid receptor activation is also involved in tramadol-induced antinociception (see DOI:10.1371/journal.pone.0125147).

Answer: As recommended, we insert the reference and rewrite the sentence to point out the opioid receptor activation by tramadol (lines 123-124, page 7).

Question 3: Line 87: it seems to be unlikely that “the six family members” indicate TRPC, TRPV, TRPM, TRPP, TRPML and TRPA. Therefore, this sentence should be revised.

Answer: As recommended, we revised the sentence, and the complete description of the abbreviations was added to the abbreviation list (lines 81-86 and 148-150, page 8).

Question 4: The contents of Figs. 1-3 partially overlap. It may be better to omit or reduce some of the images shown in Fig. 1 and provide a diagram showing the chemical structure of TRPV4 channels mentioned in the second paragraph on page 3.

Answer: Thanks for your comment, but in figure 1 we showed the TRPV4 agonists that induce nociception and the different locations where this receptor is expressed, so we changed the figure, but we can exclude it if necessary (Page 11).

Question 5: Figure 1: correspondence between large circle and small circle is not clear. Small circle does not seem to be located in the DRG. Judging from the colors used, TRPV4 seems to be expressed in spinal cord, DRG neuron fiber and peripheral neuron fiber. Is this OK? Please reply to this question.

Answer: As recommended, we changed figure 1, and the TRPV4 receptor is expressed in the spinal cord, DRG neuron fiber, and peripheral neuron fiber, as shown in figure 1 (Page 11).

Question 6. Figure 2: (a), (b), ... in its legend do not correspond to (A), (B), … in Fig. 2. Please amend this point. What is the figure shown in the upper left of Fig. 2? Please reply to this question.

Answer: We appreciate your recommendation, we checked the legend and changed it as suggested, also, in the upper left side of figure 2 we showed an example of spinal cord surgery, so we changed figure 2 to be clearer (Page 22, lines 428-438).

Question 7. Figure 3: (a), (b), ... in its legend do not correspond to (A), (B), … in Fig. 3. Please amend this point. In its legend, please explain shortly the figure (injection method) shown in the upper left of Fig. 3.

Answer: We appreciate your recommendation, we checked the legend and changed it as suggested, also, in the upper left side of figure 3 we showed an example of spinal cord surgery, so we changed figure 3 to be clearer (Page 33, lines 428-438).

Question 8. Tables 1-3: there should be more space between each item. Using “SD” instead of “Sprague-Dawley” might save space in each Table. “Author” should be “Reference”. It is not necessary to define “TRPV4” twice (title and footnote). Please amend this point.

Answer: As recommended, we changed the tables (Table 1 to 3).

Specific points:

Point 1. Lines 8 and 9: “role” has been written twice. Please amend this point.

Answer: As requested, we changed this point (lines 16-17, page 2).

Point 2. Line 13: “(NO)” should be “(NO),”.

Answer: As requested, we changed this point (line 22, page 2).

Point 3. Line 22: not “TRPV1” but “TRPV4”? Please check about this matter.

Answer: Thanks for the suggestion, but it is right the keywords because TRPV1 is co-express with TRPV4, and we did not mention it in the title and the abstract. The TRPV4 is already mentioned in the title.

Point 4. Line 68: “are” should be “is”.

Answer: As requested, we changed this word (line 143, page 8).

Point 5. Line 87: it may be better for “TRPV1/4” to be changed to “TRPV1-4”.

Answer: As requested, we changed this point (lines 165-166, page 9).

Point 6. Line 125: please expand “UVB”.

Answer: As requested, we expanded “UVB” and inserted the abbreviation in the list (lines 210 and 87, pages 5 and 10).

Point 7. Lines 163 and 164: “TRPV4 .. IHC)” should be revised in English.

Answer: As requested, we corrected the English in this line (lines 260-261, page 13).

Point 8. Lines 172 and 173: this sentence should be revised.

Answer: As requested, we revised this sentence (lines 270-272, page 13).

Point 9. Line 204: “agonists” should be “agonist”.

Answer: As requested, we changed this point (line 306, page 15).

Point 10. Line 212: please move “i.t” to the correct position.

Answer: As requested, we changed this point (line 314, page15).

Point 11. Line 240: please explain “LV-NC” shortly.

Answer: As requested, we explained the LV-NC abbreviation and inserted the abbreviation in the list (lines 64 and 344-345, pages 4 and 16).

Point 12. Line 261: please write concrete drugs of “the agonists or inhibitors”.

Answer: As requested, we write the drugs used in this study (lines 368-369, page 17).

Point 13. Line 267: “DRG” should be “DRG neurons”.

Answer: As requested, we changed this point (lines 375-376, page 17).

Point 14. Line 283: what kind of “a selective antagonist”? Please make this point clear.

Answer: As requested, we insert the name of the selective antagonist (lines 393-394, page 18).

Point 15. Line 300: what kind of “an agonist”? Please make this point clear.

Answer: As requested, we insert the name of the agonist (line 412, page 19).

Point 16. Line 304: not “Thus” but “Thus,“.

Answer: As requested, we changed this point (line 417, page 19).

Point 17. The definition of CCD is inconsistent between line 181 and the footnote of Table 1. Please amend this point.

Answer: As requested, the right definition of CCD was in line 181, so we changed the footnote of Table 1 (Page 21).

Point 18. Last line on page 8, line 476 and the footnote of Table 3: not “real-time” but “real-time PCR”.

Answer: As requested, we changed this point in Tables 1-3 (Pages 21, 32, 36).

Point 19. Lines 329 and 330: is “a better treatment” “TRP channels”? Please amend this point.

Answer: In this point, the better treatment should be inhibition of the TRP channel, so we change the sentence (lines 445-446, page 23).

Point 20. Line 344: please correct English in this line.

Answer: As requested, we corrected the English in this line (lines 459-461, page 23-24).

Point 21. Line 350: please correct the position of “(SB366791)”.

Answer: As requested, we changed the position of (SB366791) (lines 467, page 24).

Point 22. Lines 351-353: it may be better to shortly mention possible cellular mechanisms for this oxytocin action.

Answer: As requested, we insert an explanation about oxytocin action in pain perception (lines 469-471, page 24).

Point 23. Lines 357-361: it may be better to shortly mention possible cellular mechanisms for this “botulinum toxin type A” action. In general, the abbreviation for “botulinum toxin type A” uses “BoNT-A” but not “BTX-A”.

Answer: As requested, we changed this point and explained the BoNT-A mechanism in pain modulation (lines 479, 480-482, page 24).

Point 24. Line 430: is “PLC inhibitor” “PKA and PKC”? Some clarification is necessary here.

Answer: As requested, we added an explanation about PLC inhibitor, PKA and PKC (lines 557-559, pages 27-28).

Point 25. Line 436: please explain here “i.g.”, although this is given in the footnote of Table 3.

Answer: As requested, we explain the abbreviation (line 571, page 28).

Point 26. Line 439: please give an expansion of “CGRP”.

Answer: As requested, we explain the abbreviation (line 575, page 28).

Point 27. Line 512: what is “de TRPV4 expression”? Please reply to this question.

Answer: It was a typing mistake, so we corrected this point (line 655, page 35).

Point 28. Line 516: not “causing” but “causes”? Please reply to this question.

Answer: Yes, as requested, we changed this point (lines 674, page 35).

Point 29. Line 593: please correct English in this line.

Answer: As requested, we corrected the English in this line (lines 755-757, page 39).

Point 30. There appear to be much more mistakes than pointed out above. Please check the manuscript very carefully.

Answer: Thanks for the suggestion, we check the manuscript.

Sincerely yours,

Professor Gabriela Trevisan

Round 2

Reviewer 1 Report

The manuscript has been notably ameliorated. There are still some sentences that, with a careful reading of the proofs, can be completed and/or corrected. The English language requires in fact some minor spell check and and/or style check (one for all see for ex. the sentence in the Abstract "However, knowledge about the complete 17 mechanisms is unclear, but the role of oxidative compounds has been evaluated." where it appears that there are two adversatives ... However... but).

Author Response

Manuscript antioxidants-1994060

ANSWERS TO THE COMMENT OF REVIEWER 1

Question 1: The English language requires in fact some minor spell check and and/or style check (one for all see for ex. the sentence in the Abstract "However, knowledge about the complete 17 mechanisms is unclear, but the role of oxidative compounds has been evaluated." where it appears that there are two adversatives ... However... but).

Answer: Thanks for the suggestion, we change the sentence and send the manuscript to English revision (Line 16, page 2).

Sincerely yours,

Professor Gabriela Trevisan

Reviewer 2 Report

Although this revised manuscript has been partially amended according to my comments, there are still so many points that may serve to amend this manuscript as follows:

1.     Line 18: not “receptors” but “receptor”.

2.     Line 35: why is there not TRPV4 in Keywords?  Are TRPV1 and TRPA1 contained in TRP channels?

3.     Line 53: “Calcium ..” will be unnecessary.

4.     Line 75: “Magnesium ..” will be unnecessary.

5.     Line 78: not “iNOS” but “NOS”.

6.     Line 87: “Sodium ..” will be unnecessary.

7.     Line 90: “Transient ..” will be unnecessary.

8.     Line 96: the definition of “TRPV” but not “TRPV1” should be given.

9.     Line 98: this should be put in the last of Abbreviations.

10.  Lines 289 and 290: are “peripheral nociceptive neurons” a part of “dorsal root ganglion neurons”?  It may be better to revise this sentence.

11.  Line 299: not “mediate” but “mediates”.

12.  Line 441: there is no explanation about “I-κB”.  Please amend this point.

13.  Line 461: is “neuropathic-induced” OK?  Please amend this point.

14.  Line 488: “HEK-TRPV4 cells” is an inappropriate expression.  Please amend this point.

15.  Line 506: it is not clear how the content stated in lines 502-506 is related to “Table 1 and Figure 2”.  Please make this point clear.

16.  Line 530: the content stated in lines 529 and 530 does not seem to be shown in Table 1.  Please make this point clear.

17.  Line 536: the content stated in lines 534-536 does not seem to be shown in Table 1.  Please make this point clear.

18.  The end of Table 1: Zhang (2019) is not [62].  The authors should check all references.

19.  The legend of Figure 2: (A) does not show “Mechanism”.  What is (C-e)?  The authors should revise this legend.

20.  Line 587: not “insulin growth factor” but “insulin-like growth factor”?

21.  Line 591: “are” should be “is”.

22.  Lines 593 and 594: a reference should be put here.

23.  Lines 600-602: does “this compound” mean “oxytocin receptor”?  This sentence seems to make no sense.  Please revise this sentence.

24.  Lines 611 and 612: not “inhibit” but “inhibits”; not “modulate” but “modulates”.

25.  Line 620: the content stated in lines 619 and 620 does not seem to be shown in Table 1.  Please make this point clear.

26.  Line 638: “hypotonic saline (a TRPV4 agonist)” is an inappropriate expression.  Please amend this point.

27.  Lines 649 and 650: what were the results obtained from Ca2+ imaging experiments?  There seems to be no description of the results.  Please amend this point.

28.  Line 658: “Trpv4-/-“ should be “Trpv4-/- mice”.

29.  Line 663: “YEEIP” should be explained in this line but not lines 901 and 902.

30.  Line 684: what is induced by paclitaxel?  Please amend this line.

31.  Line 690: is “as” OK?  Is “using” OK?  Please check English.

32.  Line 707: it is not clear what are “these antagonists”.  Please amend this point.

33.  Line 722: English in this line should be revised.

34.  Table 2: not “1mg/ml” but “1 mg/ml”.

35.  Line 824: not “increase” but “increases”.

36.  The legend of Figure 3: (A) does not show “Mechanisms” (see line 821).  What occurs after calcium influx (see line 829)?  Reply to this question.

37.  The descriptions in lines 794-818 and 831-837 are duplicated.  Please amend this point.

38.  Lines 845 and 846: not “receptor” but “channel”?; there are two “expression”.  Please amend these points.

39.  Footnote in Table 3: where is the explanation of “ODN”?

40.  Line 906: not “transduce” but “transduces”?

41.  Line 929: “.. decreased these ..” should be revised.

42.  There appear to be much more mistakes than pointed out above.  Please check the manuscript very carefully.

Author Response

Manuscript antioxidants-1994060

ANSWERS TO THE COMMENT OF REVIEWER 2

We want to thank you very much for our manuscript's essential and constructive suggestions, and they will undoubtedly improve its scientific value.

Question 1: Line 18: not “receptors” but “receptor”.

Answer: As requested, we change this point (Line 18, page 2).

Question 2: Line 35: why is there not TRPV4 in Keywords? Are TRPV1 and TRPA1 contained in TRP channels?

Answer: We did not use the TRPV4 in keywords because the TRPV4 is already mentioned in the title. TRPV1 and TRPA1 are members of the TRPs channels family and are co-express with TRPV4. In addition, some studies evaluated the TRPV4 involvement in neuropathic pain associated with TRPV1 or TRPA1 activation. But if

Question 3: Line 53: “Calcium ..” will be unnecessary.

Answer: As requested, we change this point (Page 3).

Question 4: Line 75: “Magnesium ..” will be unnecessary.

Answer: As requested, we changed this point (Page 4).

Question 5: Line 78: not “iNOS” but “NOS”.

Answer: As requested, we changed this point (Line 67, page 4).

Question 6: Line 87: “Sodium ..” will be unnecessary.

Answer: As requested, we changed this point (Page 4).

Question 7: Line 90: “Transient ..” will be unnecessary.

Answer: As requested, we change this point (Page 4).

Question 8: Line 96: the definition of “TRPV” but not “TRPV1” should be given.

Answer: As requested, we changed this point (Line 83, page 4).

Question 9: Line 98: this should be put in the last of Abbreviations.

Answer: As requested, we changed this point (Line 87, page 4).

Question 10: Lines 289 and 290: are "peripheral nociceptive neurons" a part of "dorsal root ganglion neurons"? It may be better to revise this sentence.

Answer: Thanks for your question, but they are different structures, so we revised the sentence (Lines 221-222, page 10).

Question 11: Line 299: not “mediate” but “mediates”.

Answer: As requested, we changed this point (Line 224, page 12).

Question 12: Line 441: there is no explanation about “I-κB". Please amend this point.

Answer: As requested, we explained this abbreviation (Line 332, page 16).

Question 13: Line 461: is “neuropathic-induced" OK? Please amend this point.

Answer: As requested, we changed this point (Lines 348-349, page 16).

Question 14: Line 488: “HEK-TRPV4 cells” is an inappropriate expression. Please amend this point.

Answer: As requested, we changed this point (Line 375, page 17).

Question 15: Line 506: it is not clear how the content stated in lines 502-506 is related to "Table 1 and Figure 2". Please make this point clear.

Answer: In this line, we intend to summarize the results of the CCD model observed in Table 1 and Figure 2, so we revised those sentences (Lines 391-396, page 17).

Question 16: Line 530: the content stated in lines 529 and 530 does not seem to be shown in Table 1. Please make this point clear.

Answer: As requested, we revised the Table 1 citation in these lines and just cited the table when we pointed out the antagonist that we mentioned in the table (Line 415, page 18).

Question 17: Line 536: the content stated in lines 534-536 does not seem to be shown in Table 1. Please make this point clear.

Answer: As requested, we revised the Table 1 citation (Page 21).

Question 18: The end of Table 1: Zhang (2019) is not [62]. The authors should check all references.

Answer: As requested, we checked all references in the three tables (Table 1 to 3).

Question 19: The legend of Figure 2: (A) does not show “Mechanism”. What is (C-e)? The authors should revise this legend.

Answer: As requested, we changed this point and corrected the typing mistake in the legend (Lines 425, page 21)

Question 20: Line 587: not “insulin growth factor” but “insulin-like growth factor”?

Answer: As requested, we changed this point (Line 453, page 22).

Question 21: Line 591: “are” should be “is”.

Answer: As requested, we changed this point (Line 460, page 24).

Question 22: Lines 593 and 594: a reference should be put here.

Answer: As requested, we inserted a reference (Line 463, page 24).

Question 23: Lines 600-602: does “this compound” mean “oxytocin receptor"? This sentence seems to make no sense. Please revise this sentence.

Answer: As requested, we revised this sentence (Lines 469-472, page 24).

Question 24: Lines 611 and 612: not “inhibit” but “inhibits”; not “modulate” but “modulates”.

Answer: As requested, we changed those points (Lines 481 and 482, page 24).

Question 25: Line 620: the content stated in lines 619 and 620 does not seem to be shown in Table 1. Please make this point clear.

Answer: As requested, we removed the Table 1 citation (Line 490, page 25).

Question 26: Line 638: “hypotonic saline (a TRPV4 agonist)" is an inappropriate expression. Please amend this point.

Answer: As requested, we changed this point (Lines 508-509, page 25).

Question 27: Lines 649 and 650: what were the results obtained from Ca2+ imaging experiments? There seems to be no description of the results. Please amend this point.

Answer: The study showed increased calcium influx after paclitaxel treatment in cultured nociceptors, so we described these results (Lines 520-521, page 26).

Question 28: Line 658: “Trpv4-/-“ should be “Trpv4-/- mice”.

Answer: As requested, we changed this point (Line 529, page 26).

Question 29: Line 663: “YEEIP” should be explained in this line but not lines 901 and 902.

Answer: As requested, we changed the position of YEEIP definition (Line 534, page 26).

Question 30: Line 684: what is induced by paclitaxel? Please amend this line.

Answer: After paclitaxel injection, the neuropathic pain model is established (Line 555, page 27).

Question 31: Line 690: is "as" OK? Is "using" OK? Please check English.

Answer: As requested, we checked the English (Lines 561 and 562, page 27)

Question 32: Line 707: it is not clear what are "these antagonists". Please amend this point.

Answer: As requested, we changed the sentence, these antagonists refer to TRPV4 and TRPA1 antagonists (Lines 578-579, page 28).

Question 33: Line 722: English in this line should be revised.

Answer: As requested, we checked the English (Lines 592-594, page 29).

Question 34: Table 2: not “1mg/ml” but “1 mg/ml”.

Answer: As requested, we corrected this point (Table 2).

Question 35: Line 824: not “increase” but “increases”.

Answer: As requested, we corrected this point (Line 670, page 33).

Question 36: The legend of Figure 3: (A) does not show "Mechanisms" (see line 821). What occurs after calcium influx (see line 829)? Reply to this question.

Answer: As requested, we corrected this point, and after calcium influx is observed neuropathic pain (Lines 671and 679, pages 34-35).

Question 37: The descriptions in lines 794-818 and 831-837 are duplicated. Please amend this point.

Answer: Yes, it was a typing mistake, so we deleted the duplicated lines (Page 34).

Question 38: Lines 845 and 846: not “receptor” but "channel"?; there are two "expression". Please amend these points.

Answer: Yes, the TRP channels also can be named TRP receptors, but we changed the word as suggested (Lines 688, page 35).

Question 39: Footnote in Table 3: where is the explanation of “ODN”?

Answer: Thanks for the suggestion, we checked the manuscript and added the explanation of ODN in all tables footnotes (Table 1 to 3).

Question 40: Line 906: not “transduce” but “transduces”?

Answer: As requested, we corrected this point (Line 731, page 38).

Question 41: Line 929: “.. decreased these ..” should be revised.

Answer: As requested, we corrected this point (Line 752, page 38).

Question 42: There appear to be much more mistakes than pointed out above. Please check the manuscript very carefully

Answer: Thanks for the suggestion, we change the sentence and send the manuscript to English revision.

Sincerely yours,

Professor Gabriela Trevisan

Round 3

Reviewer 2 Report

Although this re-revised manuscript is partly amended according to my comments, there are still many points that should be amended, as stated in the following.  It should be clear in the figures where TRPV4 is activated.

1.     Line 61: not “i.g” but “i.g.”.  Please see line 574.

2.     Line 63: “i.v.” will be better than “i.v”, although both of them are used in Table 3.

3.     Line 77: not “s.c” but “s.c.”.  Please see Table 1.

4.     Line 106: “pain” following “peripheral”?

5.     Line 120: why “SSNRI” is not used here?  Please see lines 115 and 116.

6.     Line 121: not “showed” but “show”?

7.     Line 133: not “act” but “acts”?

8.     Line 135: is it OK to use “modulators” twice?

9.     Line 137: is “neuropathic pain” “acute disease”?

10.  Line 150: does “magnesium” influx occur under physiological conditions?

11.  Lines 151, 153, 184, 432, 660 and Fig. 1: not “receptor” but “channel”?

12.  Line 223: are “dorsal root ganglion neuron” and “peripheral nociceptive neuron” different types of neurons?  Please make this point clear.

13.  Line 257: not “expression” but “expressions”

14.  Line 290: is “compressed the DRG” OK?

15.  Lines 296 and 339, and Table 1: not “i.t” but “i.t.”.

16.  Line 431: “spinal dorsal horn” will be better than “dorsal spinal cord”.

17.  Line 435: “the” should be deleted.

18.  Line 464: is “threshold was blocked” OK?

19.  Lines 560-567: “PLC” is used in line 560 before “PLC” is defined in lines 566 and 567.  Please amend this point.

20.  Line 573: “HC-030031” is a TRPA1 antagonist.  Please amend this point.

21.  Line 666: is “model-induced” OK?

22.  Lines 671-673: Fig. 3A does not show CINP.  This sentence should be revised.

23.  There are several questions about the figures given.  It is unknown what “peripheral nociceptive neuron” means.  Please note that the central terminal of DRG neuron is located in the spinal dorsal horn, while the peripheral terminal resides in the skin.  Is TRPV4 activated in the middle of nerve fibers?  It is unclear where TRPV4 in Fig. 2 is located.  Does Fig. 3 show that TRPV4 is activated in the cell body of DRG neuron?

Author Response

Manuscript antioxidants-1994060

ANSWERS TO THE COMMENT OF REVIEWER 2

Question 1: Line 61: not “i.g” but “i.g.”.  Please see line 574.

Answer: As requested, we changed this point and checked all the manuscript (lines 61, pages 4).

Question 2: Line 63: “i.v.” will be better than “i.v”, although both of them are used in Table 3.

Answer: As requested, we change this point and checked all the manuscript (line 63, page 4).

Question 3: Line 77: not “s.c” but “s.c.”.  Please see Table 1.

Answer: As requested, we changed this point and checked all the manuscript (line 77, page 4).

Question 4: Line 106: “pain” following “peripheral”?

Answer: Here we want to said the difference between central pain and peripheral pain, so we changed the sentence (Line 105-106, page 6).

Question 5: Line 120: why “SSNRI” is not used here?  Please see lines 115 and 116.

Answer: As requested, we change this point (line 120, page 7).

Question 6:     Line 121: not “showed” but “show”?

Answer: As requested, we changed this point (Line 120, page 7).

Question 7:      Line 133: not “act” but “acts”?

Answer: As requested, we change this point (Line 132, page 7).

Question 8:      Line 135: is it OK to use “modulators” twice?

Answer: Yes, in this case we cannot use another word to describe this.

Question 9: Line 137: is “neuropathic pain” “acute disease”?

Answer: As requested, we changed this point (Line 136, page 7).

Question 10: Line 150: does “magnesium” influx occur under physiological conditions?

Answer: The ions influx occurred after TRP activation, but we changed the sentence (Lines 148, page 8).

Question 11:  Lines 151, 153, 184, 432, 660 and Fig. 1: not “receptor” but “channel”?

Answer: As requested, we change this point and checked all the manuscript (line 150, 152, 155, 183, 431, 659 page 8, 9, 22, 33).

Question 12. Line 223: are “dorsal root ganglion neuron” and “peripheral nociceptive neuron” different types of neurons?  Please make this point clear.

Answer: Yes, dorsal root ganglion neuron and peripheral nociceptive neuron are different types of neurons (Lines 222-223, page 11)

Question 13.  Line 257: not “expression” but “expressions”

Answer: As requested, we changed this point (line 257, page 13).

Question 14.  Line 290: is “compressed the DRG” OK?

Answer: As requested, we changed this point (line 290, page 14).

Question 15.  Lines 296 and 339, and Table 1: not “i.t” but “i.t.”.

Answer: As requested, we changed this point and checked all the manuscript (lines 296, 339, and Table 1 pages 14, 16).

Question 16:  Line 431: “spinal dorsal horn” will be better than “dorsal spinal cord”.

Answer: As requested, we changed this point (Line 431, page 22).

Question 17:  Line 435: “the” should be deleted.

Answer: As requested, we changed this point (line 434, page 22).

Question 18:  Line 464: is “threshold was blocked” OK?

Answer: As requested, we changed this point (Line 465, page 24).

Question 19.  Lines 560-567: “PLC” is used in line 560 before “PLC” is defined in lines 566 and 567.  Please amend this point.

Answer: As requested, we changed this point (line 561, page 28).

Question 20.  Line 573: “HC-030031” is a TRPA1 antagonist.  Please amend this point.

Answer: Yes, it was a typing mistake, here we explained the co-administration of TRPV4 and TRAP1 antagonists (line 574, page 28).

Question 21.  Line 666: is “model-induced” OK?

Answer: As requested, we changed this point (line 665, page 34).

Question 22.  Lines 671-673: Fig. 3A does not show CINP.  This sentence should be revised.

Answer: As requested, we change this sentence (line 671, page 34).

Question 23.  There are several questions about the figures given.  It is unknown what “peripheral nociceptive neuron” means.  Please note that the central terminal of DRG neuron is located in the spinal dorsal horn, while the peripheral terminal resides in the skin.  Is TRPV4 activated in the middle of nerve fibers?  It is unclear where TRPV4 in Fig. 2 is located.  Does Fig. 3 show that TRPV4 is activated in the cell body of DRG neuron?

Answer: Thanks for your question, we changed the figures as suggested.

We also send the manuscript to English revision, in attached the certificate.

Sincerely yours,

Professor Gabriela Trevisan
